# IMPROVING LOW-BIT POST TRAINING QUANTIZATION: A DATA-FREE APPROACH

## ABSTRACT

Post-training quantization (PTQ) without access to real data is enabling efficient model optimization and deployment in scenarios where privacy or proprietary constraints restrict the use of original datasets. Traditional data free quantization methods rely on Batch Normalization (BN) statistics from the trained full-precision model to generate calibration dataset for quantization. However, this reliance on BN statistics limits their applicability to deep neural networks (DNNs) without BN layer such as AlexNet. In this paper, we propose a calibration dataset generation algorithm that is agnostic to BN statistics, leveraging just the back-propagation to create synthetic images for PTQ. We also demonstrate that it is not necessary to include samples from every target category in the calibration dataset to get the representative activation ranges for quantization. Extensive experiments with both large and lightweight models on large-scale image classification tasks demonstrate that our method consistently improves quantization performance across various DNN architectures, especially in low-bit settings. Notably, in 4-bit quantization, we achieve an improvement of 3.42% in top-1 accuracy for the ResNet18 model and 3.14% for the InceptionV3 model compared to the state-of-the-art (SOTA) DSG method. Importantly, we use very few synthetic samples for quantization compared to other methods.

## 1 INTRODUCTION

DNNs have achieved remarkable success across a wide range of applications, driven by advancements in computational power, large-scale datasets, and innovative algorithms (LeCun et al., 2015). These networks have revolutionized the field of computer vision. They power image classification (Krizhevsky et al., 2012), object detection (Girshick et al., 2014), and semantic segmentation (Long et al., 2015) systems with unprecedented accuracy. With these domains, DNNs have made significant strides in robotics, enhancing the ability of robots to perceive and interact with their environments autonomously (Mnih et al., 2015). In autonomous driving, deep learning models play a crucial role in enabling vehicles to understand and navigate complex road conditions safely (Bojarski et al., 2016). The medical field has also seen transformative impacts, with DNNs aiding in disease diagnosis, medical imaging analysis, and personalized treatment planning (Esteva et al., 2017). However, deploying these neural networks on resource-constrained devices remains a considerable challenge due to their substantial memory requirements and intensive computational demands (Howard, 2017). The advancement of hardware capable of low-precision computations has made quantization a favored method for addressing these challenges (Jacob et al., 2018). Quantizing the floating-point values of weights and/or activations in a neural network to integers can significantly reduce the model size without altering the architecture.

Quantization methods are generally categorized into two types: quantization-aware training (QAT) and PTQ. QAT incorporates quantization into the training process, allowing the model to learn and adapt to the quantized weights and activations. This approach typically results in higher accuracy compared to PTQ, as the model is optimized to perform well under the constraints of quantization (Choi et al., 2018). While the performance degradation is minimal with QAT, the process can be computationally intensive and time-consuming (Nahshan et al., 2021). PTQ involves applying quantization techniques to a pre-trained model without training process (Cai et al., 2020; Qin et al., 2023). One of the challenges in quantization is determining the range of activation values in DNN. Many quantization methods are designed for data-driven scenarios, requiring access to either training or

validation data, or relying on a small set of unlabeled calibration data to accurately represent the activation values (Banner et al., 2019). However, real data may not always be available due to some constraints, particularly with proprietary data. To mitigate this, data-free quantization techniques have been developed to allow for the quantization without needing access to real data (Cai et al., 2020). Among existing data-free quantization methods, generative approaches create calibration data by aligning the distribution of synthetic data with the BN statistics of the trained full-precision model (Xu et al., 2020). Generative approaches can greatly reduce the accuracy gap between quantized models that use synthetic data and those that use real data (Qin et al., 2023; Cai et al., 2020). However, relying on BN statistics limits the applicability of these methods to neural networks which don't have BN layer such as AlexNet (Krizhevsky et al., 2012). In our work, we demonstrate that it is not necessary to depend on the BN statistics of the full-precision model to generate a calibration dataset for achieving optimal performance from the quantized model. Additionally, the common practice when curating a calibration dataset is to include at least one image from each target class to calibrate the image classification model (Zhang et al., 2023). We show that optimal performance can be achieved without having a sample from each category and that fewer samples can suffice.

In this paper, we primarily explore the method of generating synthetic calibration dataset for PTQ of DNN models. We demonstrate that the BN statistics of the original floating-point model are not necessary to create effective synthetic data. The main contributions of this paper are:

- We propose a method to generate synthetic data using the trained full-precision model agnostic to BN statistics, making our approach applicable to any model architecture.

- We experimentally demonstrate that it is not necessary to include a sample from each target category in the calibration dataset; selecting only a few target classes is sufficient to create an effective calibration dataset which in turn reduces PTQ time.

- Through extensive PTQ comparisons on several standard network architectures such as ResNet18/20/50 (He et al., 2016), SqueezeNext (Gholami et al., 2018), InceptionV3 (Szegedy et al., 2016), and lightweight architectures like ShuffleNet (Zhang et al., 2018), and MobileNetV2/V3 (Sandler et al., 2018; Howard et al., 2019), we show that our method significantly outperforms existing generative quantization methods with calibration. Specifically, in 4-bit precision setting on the ResNet18 and InceptionV3 models, our method improves the top-1 accuracy by over 3.42% and 3.14% compared to the SOTA DSG method.

## 2 RELATED WORK

In this section, we first review and organize existing research on quantization into two primary methodologies: QAT and PTQ. Additionally, we categorize these methods based on whether they require any form of training or validation data for the quantization process.

### 2.1 QUANTIZATION AWARE TRAINING

QAT is an advanced technique in which the quantization process is integrated into the training phase of the neural network. (Jacob et al., 2018) integrate low-precision computations typically used during inference into the forward pass of training, while maintaining standard backpropagation with floating-point weights and biases to allow precise adjustments. This approach allows the model to learn and adapt to the quantization noise, resulting in higher accuracy for the quantized model compared to PTQ. (Courbariaux et al., 2016) introduce the concept of binarized neural networks (BNNs), where both weights and activations are constrained to +1 or -1. They replace most arithmetic operations with bit-wise operations to reduce memory size. (Krishnamoorthi, 2018) evaluated various quantization methods and bit-widths, revealing that even with per-channel quantization, lightweight networks do not reach baseline accuracy with INT8 PTQ and require QAT. Using an annealing learning rate schedule and a very small final learning rate, (McKinstry et al., 2018) show that many networks can be fine-tuned for just one epoch after quantizing to INT8 and still reach baseline accuracy. (Zhou et al., 2016) introduce DoReFa-Net, a framework for training neural networks with low bit-width weights, activations, and gradients. They also propose QAT techniques to ensure effective learning with low precision. (Rastegari et al., 2016) presented XNOR-Net, a QAT-trained binary CNN architecture, achieving impressive accuracy on ImageNet and demonstrating

the feasibility of highly efficient binary networks for real-world tasks. PACT (Choi et al., 2018) dynamically adjusts the clipping values to minimize quantization error, leading to higher accuracy, especially for low-bit width quantization.

QAT maintains quantized model accuracy close to the full precision model while significantly reducing the model size and computational complexity. However, it requires more computational resources and time due to additional quantization operations, making it more complex and resource-intensive than standard training (Nahshan et al., 2021). Additionally, QAT implementation is challenging, needing modifications to the training pipeline.

## 2.2 Post Training Quantization

PTQ is a widely used technique aimed at reducing the memory and computational requirements of neural networks without the need for retraining. Unlike QAT, which incorporates quantization into the training process, PTQ is applied after the model has been fully trained using a small calibration dataset created from the original training or validation dataset. To quantize activations in DNNs, it is essential to determine the activation ranges. (Banner et al., 2019) address this by analytically finding the activation clipping ranges. They demonstrate that the activations typically follow bell-curve statistics, fitting either Laplace or Gaussian distributions, and they formulate the clipping process as an optimization problem. (Zhao et al., 2019) introduce Outlier Channel Splitting (OCS), a technique designed to handle outliers in the distribution of weights and activations that can negatively impact quantization. OCS works by identifying a small number of channels containing outliers, duplicating them, and then halving the values in those channels, which moves the affected outliers toward the center of the distribution. (Li et al., 2021) introduce a novel method called Block Reconstruction Quantization (BRECQ), which focuses on optimizing the quantization of neural network weights in a block-wise manner. BRECQ achieves this through local optimization and layer-wise error correction, significantly reducing quantization-induced errors without requiring retraining. (Nagel et al., 2020) present a novel adaptive rounding technique, AdaRound, that dynamically determines whether to round weights and activations up or down during quantization. This optimization-based approach minimizes the quantization error and preserves model accuracy without requiring retraining. Unlike AdaRound, which confines quantized weights to be within ±1 of their rounded values, AdaQuant (Hubara et al., 2021) grants more freedom. It achieves this by independently optimizing each layer's weights and quantization parameters, using the calibration set to minimize the mean-squared-error between the original and quantized layer outputs. (Cai et al., 2020) introduces a zero-shot quantization approach that eliminates the need for real calibration data by generating synthetic data. This synthetic data is used to calibrate the quantization parameters, allowing for effective quantization without access to real data. (Qin et al., 2023) addresses the homogenization of synthetic data for quantization and improves quantized model performance.

## 2.3 Data-dependent Quantization

Data-dependent quantizations are the techniques that leverage the characteristics and distribution of the training data to enhance the efficiency and accuracy of quantized neural networks. Different fixed precision data-dependent QAT schemes have been proposed in the literature. (Gupta et al., 2015) use 16 bits word length for weights, biases, and other intermediate variables during training with stochastic rounding. (Wang et al., 2019) mine channel-wise interaction to learn interacted bit count to prevent performance degradation in binary neural networks. Unlike traditional methods (Zhuang et al., 2021) first optimized the network with quantized weights and then with quantized activations. To address the increased training time and computational costs of data-driven QAT methods, PTQ methods using real data have been proposed. (Banner et al., 2018) introduce Analytical Clipping for Integer Quantization (ACIQ) to optimize activation clipping, a per-channel bit allocation policy to minimize mean-squared-error, and a bias-correction method to address inherent biases in quantized weights. This technique enhances the accuracy and efficiency of 4-bit quantized convolutional neural networks. A low-bit precision linear quantization framework is proposed by (Choukroun et al., 2019) in which the optimal solution to the quantization problem is found via mean-squared-error optimization with fixed precision constraints. (Zhao et al., 2019) sampled activation distribution using 512 training images to determine quantization grid points, then use this grid during testing. Generally, when quantizing the DNNs, assigning each floating-point weight to its nearest fixed-point value is the common approach but (Nagel et al., 2020) show that this isn't the best approach.

In AdaRound they use a better weight-rounding mechanism for PTQ that adapts to the data and the task loss. Even though these methods provide good quantization techniques, they still need access to training or validation data.

## 2.4 DATA-INDEPENDENT QUANTIZATION

Due to privacy or proprietary constraints, access to training or validation datasets is not always possible. Consequently, several works have been proposed to perform quantization using synthetic data to determine activation ranges. DFQ by (Nagel et al., 2019) leverages the scale-equivariance property of activation functions to equalize weight ranges within the network and also corrects quantization-induced biases in the error. DFQ works fine on 8-bit quantization, but the accuracy drops significantly for lower bits. (Xu et al., 2020) use a generator to generate the synthetic data and fine-tune the quantized model. Unlike other generative data-free quantization approaches, (Qian et al., 2023) determine if the knowledge carried out by generated samples is informative or not to the quantized model. They generate samples with adaptive ability to boost the quantized model performance. (Choi et al., 2020) propose an adversarial knowledge distillation framework that minimizes the possible loss for a worst case via adversarial learning by constraining a generator in the adversarial learning framework. The Inception scheme is proposed by (Haroush et al., 2020) to generate the data; they proposed knowledge distillation-based methods to improve accuracy. Methods like (Haroush et al., 2020; Cai et al., 2020; Qin et al., 2023) purely use BN statistics of the floating point model to create synthetic data. (Qin et al., 2023) show that the synthetic data generated by constraining BN statistics suffers serious homogenization at both the distribution level and sample level and improves it by slacking distribution alignment and layerwise sample enhancement. While the aforementioned methods employ various techniques to generate synthetic data for quantization and enhance model accuracy, they typically require fine-tuning or rely on BN statistics from the full-precision model, which we demonstrate is not necessary for achieving good performance.

## 3 MOTIVATION

BN is a commonly utilized technique during the training of neural networks to stabilize and accelerate the convergence process by normalizing activations within each layer. However, when it comes to generating synthetic data for PTQ, the primary objective shifts to synthesizing inputs that produce a wide and representative range of activations within the network. This objective can be effectively met through direct optimization techniques such as backpropagation, which iteratively refines synthetic inputs using the model's gradients. By doing so, we can ensure that these synthetic inputs accurately represent the target classes without the need for BN statistics. Furthermore, Qin et al. (2023) demonstrate that samples generated solely from BN statistics exhibit minimal statistical variation compared to real-world samples, which are inherently diverse. To address this, we introduce Gaussian noise during the optimization process to enhance the variability and diversity of the generated synthetic data.

Additionally, while generating representative datasets for quantization, the prevailing practice has been to select at least one sample from each class to ensure comprehensive coverage of the activation space (Zhang et al., 2023). This approach is based on the assumption that including every class is necessary to capture the full range of activations required for effective quantization. However, this method can be computationally intensive and time-consuming, especially for large datasets with numerous classes. (Banner et al., 2018) show that the statistical distributions of weights and activations follow a bell curve, indicating that large values occur very rarely compared to small values. Motivated by this for a more efficient process, we investigate whether a smaller subset of classes could provide an effective activation range. By focusing on a reduced number of target classes, our goal was to streamline the calibration data generation process while maintaining the efficacy of quantization. This exploration was driven by the hypothesis that a subset of classes could still encompass the diverse activation patterns needed for accurate model quantization as described in subsection 5.4.

---

**Algorithm 1** Generate Calibration Dataset

---

**Require:** Full precision model $F$, total number of classes $N$, number of target classes $M$, iterations $T$, learning rate $\alpha$, threshold $\epsilon$
**Ensure:** Synthetic sample $\mathbf{x}$ for each selected class
 1: Randomly select $M$ unique target classes from $N$ total classes
 2: Initialize an empty array $\mathbf{X}$ to store synthetic samples
 3: Define loss function $\mathcal{L}$
 4: **for** each target class $c$ in $M$ **do**
 5:     Initialize $\mathbf{x} \sim \mathcal{N}(0,1)$
 6:     Set target label $y = c$
 7:     **for** $t = 1$ to $T$ **do**
 8:         $\mathbf{x} \leftarrow \mathbf{x} - \alpha \nabla_{\mathbf{x}} \mathcal{L}(F(\mathbf{x}), y)$
 9:         $\mathbf{x} \leftarrow \mathbf{x} + \mathcal{N}(0, 0.1)$
10:         **if** $F(\mathbf{x}) = y$ **and** $\mathcal{L}(F(\mathbf{x}), y) < \epsilon$ **then**
11:             break
12:         **end if**
13:     **end for**
14:     Append $\mathbf{x}$ to $\mathbf{X}$
15: **end for**
16: **return** $\mathbf{X}$

---

## 4 METHOD

The algorithm 1 for generating synthetic images begins by randomly selecting $M$ unique target classes from a total of $N$ available classes, ensuring diversity in the synthetic dataset. A loss function, $\mathcal{L}$, is defined to measure the discrepancy between the model's prediction and the target label. For each selected target class, the process initiates by creating a synthetic image, $\mathbf{x}$, with values drawn from a normal distribution, $\mathcal{N}(0,1)$. The target label, $y$, is set to the current class label $c$. Inspired by the Fast Gradient Sign Method (FGSM) introduced by (Goodfellow et al., 2014), the algorithm then enters an iterative loop for a specified number of iterations, $T$. In each iteration, the synthetic image $\mathbf{x}$ is updated by performing gradient descent, where the gradient of the loss function with respect to $\mathbf{x}$ is subtracted, scaled by a learning rate $\alpha$ as shown in equation 1. This step is analogous to the FGSM approach, which perturbs the input in the direction of the gradient to maximize the loss. To introduce variability in the samples, Gaussian noise, $\mathcal{N}(0, 0.1)$, is added to the image. The algorithm checks if the model's prediction for $\mathbf{x}$ matches the target label, and if the loss falls below a predefined threshold, $\epsilon$. If both conditions are satisfied, the loop breaks early, indicating that a satisfactory synthetic image has been generated. This process is repeated for each of the $M$ target classes, and the resulting synthetic images are stored in an array $\mathbf{X}$. Finally, the array of synthetic images is returned as the output.

$$\mathbf{x} = \mathbf{x} - \alpha \nabla_{\mathbf{x}} \mathcal{L}(F(\mathbf{x}), y) \tag{1}$$

## 5 EXPERIMENTS

In this section, we perform an in-depth assessment of the performance of our approach on image classification tasks using a series of comprehensive experiments.

### 5.1 IMPLEMENTATION DETAILS

We used PyTorch (Paszke et al., 2019) to implement the proposed approach due to its robust automatic differentiation capabilities. All the experiments are performed on NVIDIA A100 GPU, and the pre-trained full precision models are obtained from pytorchcv [1]. We generate the synthetic data

---

[1] pytorchv:https://pypi.org/project/pytorchcv/

Table 1: MobileNetV2/V3 on ImageNet

| Model | Method | W-bit | A-bit | Top-1 |
|---|---|---|---|---|
| | Baseline | 32 | 32 | 72.97% |
| | SelectQ | 4 | 4 | 10.88% |
| | Ours | 4 | 4 | **11.98%** |
| MobileNetV2 | SelectQ | 6 | 6 | **70.25%** |
| | Ours | 6 | 6 | 70.12% |
| | SelectQ | 8 | 8 | 72.84% |
| | Ours | 8 | 8 | **72.87%** |
| | Baseline | 32 | 32 | 75.34% |
| | SelectQ | 4 | 4 | 0.36% |
| | Ours | 4 | 4 | **19.29%** |
| MobileNetV3 | SelectQ | 6 | 6 | 60.04% |
| | Ours | 6 | 6 | **72.84%** |
| | SelectQ | 8 | 8 | 75.04% |
| | Ours | 8 | 8 | **75.05%** |

Table 2: ResNet20 and VGG16bn on CI-FAR10

| Model | Method | W-bit | A-bit | Top-1 |
|---|---|---|---|---|
| | Baseline | 32 | 32 | 94.08% |
| | Real Data | 4 | 4 | 87.38% |
| | ZeroQ | 4 | 4 | 85.39% |
| | DSG | 4 | 4 | 87.79% |
| | Ours | 4 | 4 | **89.26%** |
| | Real Data | 6 | 6 | 93.80% |
| | ZeroQ | 6 | 6 | 93.33% |
| | DSG | 6 | 6 | 93.55% |
| ResNet20 | Ours | 6 | 6 | **93.64%** |
| | Real Data | 8 | 8 | 93.95% |
| | ZeroQ | 8 | 8 | 93.94% |
| | DSG | 8 | 8 | 93.97% |
| | Ours | 8 | 8 | **94.00%** |
| | Baseline | 32 | 32 | 93.86% |
| | Real Data | 4 | 4 | 92.50% |
| | ZeroQ | 4 | 4 | 91.79% |
| | DSG | 4 | 4 | 92.89% |
| | Ours | 4 | 4 | **93.17%** |
| | Real Data | 6 | 6 | 93.48% |
| | ZeroQ | 6 | 6 | 93.45% |
| VGG16bn | DSG | 6 | 6 | 93.68% |
| | Ours | 6 | 6 | **93.85%** |
| | Real Data | 8 | 8 | 93.59% |
| | ZeroQ | 8 | 8 | 93.53% |
| | DSG | 8 | 8 | 93.61% |
| | Ours | 8 | 8 | **93.79%** |

using algorithm 1 and use the independent calibration process from (Cai et al., 2020). In our experiments, we apply quantization to all layers and clip the activations on a per-layer basis. We utilize a Gaussian distribution to initialize the synthetic image as well as to add noise to it. For the hyperparameters, we empirically find the optimum value of total number of target labels $M$ for each model and bit setting but never go beyond 35, iterations $T$ to 100, learning rate $\alpha$ in the range [0.1, 0.2], and threshold $\epsilon$ to 0.001.

## 5.2 EVALUATION

To demonstrate the effectiveness of our approach, we evaluate it on various network architectures with different bit settings. Our experiments include AlexNet (Krizhevsky et al., 2012), VGG16bn (Simonyan & Zisserman, 2014), ResNet18/20/50 (He et al., 2016), SqueezeNext (Gholami et al., 2018), InceptionV3 (Szegedy et al., 2016), ShuffleNet (Zhang et al., 2018), and MobileNetV2/V3 (Sandler et al., 2018; Howard et al., 2019). We assess these models using various bit-width configurations, such as W4A4 (4-bit weights and 4-bit activations), W6A6, and W8A8. We use validation datasets from ImageNet (Deng et al., 2009) and CIFAR10 (Krizhevsky et al., 2009) to evaluate our approach, measuring the effectiveness by assessing the top-1 accuracy of the quantized models.

## 5.3 COMPARISON WITH SOTA METHODS

To evaluate the benefits of our proposed PTQ scheme, we compare our method with other data-free PTQ approaches, such as DSG (Qin et al., 2023), ZeroQ (Cai et al., 2020), DFQ (Nagel et al., 2019), ACIQ (Banner et al., 2018), MSE (Chen et al., 2015), KL (Sung et al., 2015), and OCS (Zhao et al., 2019), on CIFAR10 and ImageNet datasets. Notably, DSG and ZeroQ are representative generative data-free PTQ methods that reconstruct synthetic data and calibrate the quantized network. We assess these methods under various bit-width configurations, with the results presented in table 2 for the CIFAR10 dataset and table 1, 3, and 4 for the ImageNet dataset. For MobilNetV2/V3 on

ImageNet dataset we compare our method with SelectQ [2] (Zhang et al., 2023) which uses training data for quantization.

On the CIFAR10 dataset, we evaluate our method with ResNet20 and VGG16bn, as shown in table 2. Our method consistently outperforms other methods across all bit-widths. Specifically, for ResNet20, our method improved accuracy by approximately 1.47% over DSG in the 4-bit setting. For the ImageNet dataset, we conducted experiments on MobileNetV2, MobileNetV3, ResNet18/50, SqueezeNext, InceptionV3, ShuffleNet, and AlexNet models. Our method consistently outperforms other quantization methods across different bit-widths. Notably, for MobileNetV3, our method achieved a significant improvement of 18.93% over SelectQ in the 4-bit setting, reaching 19.29%. In the case of ResNet18, our method achieved the highest accuracy in the 4-bit setting, with a 3.42% improvement over DSG, reaching 43.32%. For ResNet50, our method improved accuracy by 1.50% over DSG in the 4-bit setting, achieving 57.62%. InceptionV3 showed a significant improvement with our method, achieving 60.31% in the 4-bit setting, which is 3.14% higher than DSG, while slightly improving the accuracies in 6 and 8-bit settings. In table 3 we also demonstrate the quantization results of AlexNet model using our method which is not shown by other methods due to the unavailability of BN layer in AlexNet.

Furthermore similar to methods such as ZeroQ and DSG we require a small number of synthetic samples to achieve effective PTQ. Empirically, we have determined the effective sample size for each model and bit setting. For example, we use 25 samples for ResNet18 in a 4-bit setting and 35 samples for ResNet50 in a 4-bit setting. Importantly, we never exceed 35 samples for effective quantization across all models and bit settings. This substantial reduction in sample size does not compromise quantization performance, making our method highly efficient for various classification models.

Overall, our quantization method demonstrated superior performance across various models and bit-widths with fewer samples, particularly in the 4-bit setting. It consistently achieved high accuracy, making it suitable for resource-constrained environments without significant loss in performance.

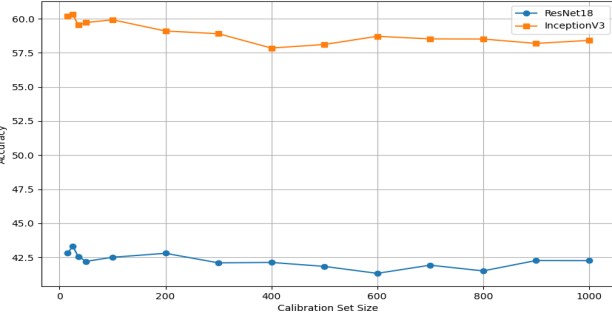

Figure 1: Quantized model performance with different calibration set size

## 5.4 Effect of Sample Size on Quantization

To evaluate the necessity of having a large sample size for calibration datasets in model quantization, we conducted two experiments using the ResNet18, InceptionV3, and ShuffleNet models.

In the first experiment, we tested different unique sample sizes ranging from 15 to 1000 and measured the top-1 accuracy of the quantized models using our synthetic dataset generation method and reported in figure 1. Contrary to the common belief that a larger calibration dataset size leads to better quantized model performance, our findings reveal that the best results are achieved with a smaller sample size. For ResNet18, the highest accuracy of 43.32% was obtained with 25 samples, while for InceptionV3, the highest accuracy of 60.32% was also achieved with 25 samples. Interestingly, increasing the sample size beyond this point resulted in a decline in accuracy. For instance,

---

[2]Importantly note that SelectQ requires access to the training data

Table 3: SqueezeNext, InceptionV3, and ShuffleNet on ImageNet

| Model | Method | W-bit | A-bit | Top-1 |
|---|---|---|---|---|
| SqueezeNext | Baseline | 32 | 32 | 69.38% |
| | Real Data | 6 | 6 | 66.51% |
| | ZeroQ | 6 | 6 | 39.83% |
| | DSG | 6 | 6 | 66.23% |
| | Ours | 6 | 6 | **66.30%** |
| | Real Data | 8 | 8 | 69.23% |
| | ZeroQ | 8 | 8 | 68.01% |
| | DSG | 8 | 8 | **69.27%** |
| | Ours | 8 | 8 | 69.21% |
| InceptionV3 | Baseline | 32 | 32 | 78.80% |
| | Real Data | 4 | 4 | 73.50% |
| | ZeroQ | 4 | 4 | 12.00% |
| | DSG | 4 | 4 | 57.17% |
| | Ours | 4 | 4 | **60.31%** |
| | Real Data | 6 | 6 | 78.59% |
| | ZeroQ | 6 | 6 | 75.14% |
| | DSG | 6 | 6 | 78.12% |
| | Ours | 6 | 6 | **78.35%** |
| | Real Data | 8 | 8 | 78.79% |
| | ZeroQ | 8 | 8 | 78.70% |
| | DSG | 8 | 8 | 78.81% |
| | Ours | 8 | 8 | **78.83%** |
| ShuffleNet | Baseline | 32 | 32 | 65.07% |
| | Real Data | 6 | 6 | 56.25% |
| | ZeroQ | 6 | 6 | 39.92% |
| | DSG | 6 | 6 | **60.71%** |
| | Ours | 6 | 6 | 60.60% |
| | Real Data | 8 | 8 | 64.52% |
| | ZeroQ | 8 | 8 | 64.46% |
| | DSG | 8 | 8 | 64.87% |
| | Ours | 8 | 8 | **64.93%** |
| AlexNet | Baseline | 32 | 32 | 59.04% |
| | Ours | 4 | 4 | 45.97% |
| | Ours | 6 | 6 | 57.22% |
| | Ours | 8 | 8 | 57.47% |

Table 4: ResNet18/50 on ImageNet

| Model | Method | W-bit | A-bit | Top-1 |
|---|---|---|---|---|
| ResNet18 | Baseline | 32 | 32 | 71.47% |
| | Real Data | 4 | 4 | 65.22% |
| | DFQ | 4 | 4 | 0.10% |
| | ACIQ | 4 | 4 | 7.19% |
| | MSE | 4 | 4 | 15.08% |
| | KL | 4 | 4 | 16.27% |
| | ZeroQ | 4 | 4 | 26.04% |
| | DSG | 4 | 4 | 39.90% |
| | Ours | 4 | 4 | **43.32%** |
| | Real Data | 6 | 6 | 71.18% |
| | ACIQ | 6 | 6 | 61.15% |
| | KL | 6 | 6 | 61.34% |
| | MSE | 6 | 6 | 66.96% |
| | DFQ | 6 | 6 | 67.30% |
| | ZeroQ | 6 | 6 | 69.74% |
| | DSG | 6 | 6 | 70.46% |
| | Ours | 6 | 6 | **70.58%** |
| | Real Data | 8 | 8 | 71.48% |
| | ACIQ | 8 | 8 | 68.78% |
| | DFQ | 8 | 8 | 69.70% |
| | KL | 8 | 8 | 70.69% |
| | MSE | 8 | 8 | 71.01% |
| | ZeroQ | 8 | 8 | 71.43% |
| | DSG | 8 | 8 | **71.49%** |
| | Ours | 8 | 8 | 71.42% |
| ResNet50 | Baseline | 32 | 32 | 77.72% |
| | Real Data | 4 | 4 | 68.13% |
| | ACIQ | 4 | 4 | 61.15% |
| | ZeroQ | 4 | 4 | 8.20% |
| | DFQ | 4 | 4 | 10.32% |
| | DSG | 4 | 4 | 56.12% |
| | Ours | 4 | 4 | **57.62%** |
| | Real Data | 6 | 6 | 76.84% |
| | ZeroQ | 6 | 6 | 75.56% |
| | DSG | 6 | 6 | 76.90% |
| | Ours | 6 | 6 | **77.08%** |
| | Real Data | 8 | 8 | 77.70% |
| | ZeroQ | 8 | 8 | 77.67% |
| | DSG | 8 | 8 | **77.72%** |
| | Ours | 8 | 8 | 77.67% |

the accuracy for ResNet18 dropped to 42.26% with 1000 samples, and for InceptionV3, it decreased to 58.41%. This counterintuitive outcome suggests that using a smaller dataset can be more effective for PTQ. Our results indicate that an optimal sample size exists, beyond which additional samples may introduce noise or redundancy, negatively impacting the quantized model's performance. These findings underscore the efficiency and effectiveness of our method, which achieves good quantization results with significantly fewer samples.

In the second experiment, we plotted the minimum and maximum activation ranges for each convolution layer in the ResNet18 and ShuffleNet models, using datasets created with 25 samples and 1000 samples, respectively. The results, illustrated in figures 2 and 3, show that the activation ranges for 25 samples are remarkably close to those obtained with 1000 samples. This observation indicates that even with a significantly smaller number of samples, the activation ranges remain stable and representative of the model's behavior. Consequently, this suggests that it is not necessary to have one sample from each class to create a representative dataset for post-training quantization.

Overall, our findings demonstrate that a smaller, more manageable dataset can be effectively used for calibration, simplifying the process and reducing the need for extensive calibration datasets.

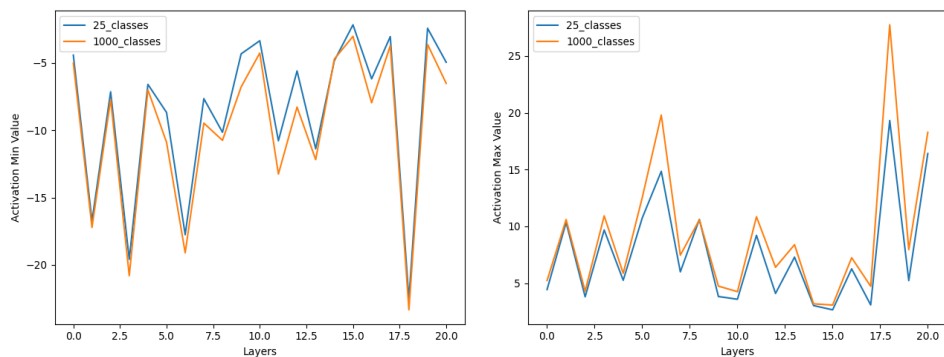

Figure 2: Minimum (left) and maximum (right) activation values for ResNet18

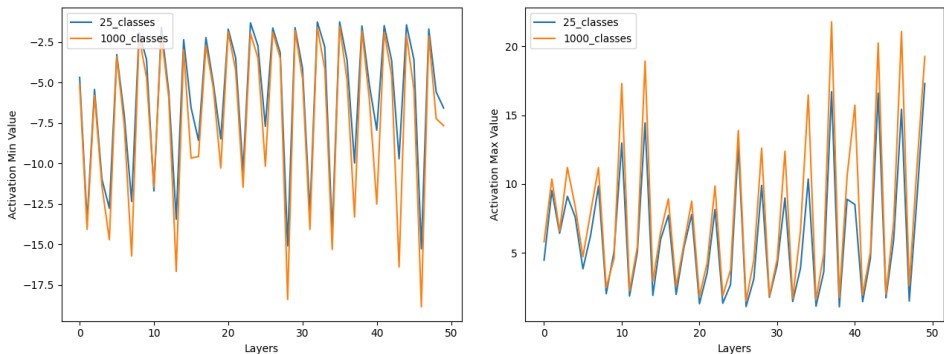

Figure 3: Minimum (left) and maximum (right) activation values for Shufflenet

## 6 CONCLUSION

In this paper, we have introduced a method for post-training data-free quantization that generates synthetic data using the trained full-precision model independent of BN statistics. This makes our approach versatile and applicable to any model architecture. Our experimental results demonstrate that it is not necessary to include samples from each target category in the calibration dataset; selecting only a few target classes is sufficient to create an effective calibration dataset. Our method consistently outperforms existing generative data-free quantization methods with calibration, as demonstrated through extensive comparisons across various standard network architectures. These include ResNet18/50, SqueezeNext, InceptionV3, as well as lightweight architectures like ShuffleNet and MobileNetV2/V3. Notably, our approach shows significant improvements in 4-bit precision settings. For instance, on the ResNet18 model, our method increases the top-1 accuracy by over 3.42% compared to SOTA DSG method. These findings underscore the effectiveness and generalizability of our approach, highlighting its potential to achieve high accuracy with lower bit-widths and fewer calibration samples. This makes it a promising solution for efficient model deployment in resource-constrained environments.

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
