# OpenReview forum: "IMPROVING LOW-BIT POST TRAINING QUANTIZATION: A DATA-FREE APPROACH"
_ICLR.cc/2025/Conference — ICLR 2025 Conference Withdrawn Submission_

### Official Review · Reviewer_gqBK · 2024-10-30

**Soundness:** 2
**Presentation:** 3
**Contribution:** 2
**Rating:** 3
**Confidence:** 4

**Summary:**

The paper proposes a data-free approach to model quantization that does not require BN information. The authors create synthetic images by backpropagating and updating noise, and the method achieves significant results on many CNN models and different datasets.

**Strengths:**

1. The paper is clearly presented and the methodology used is simple, natural and effective.
2. It looks like the authors' method achieves a nice boost at low bit quantization (W4A4) when comparing it to previous methods

**Weaknesses:**

1. **Content**: Despite the fact that 10 sides of the paper are not required under the 10 sides of the paper requirement, I still inevitably think the paper lacking in content.
2. **Experiment**: Studies still studying AlexNet in 2024 are clearly outdated, and how well do the authors' compression methods work on the most current networks: e.g., in ConvNext or ViT using LayerNorm, or SwinTransformer?
3. **Experiment**: The authors' method is missing some comparisons with more recent methods: Bai, Shipeng, et al. "Unified data-free compression: Pruning and quantization without fine-tuning." Proceedings of the IEEE/CVF International Conference on Computer Vision. 2023

I'm willing to raise my score if my concern is resolved.

**Questions:**

1. In the paper, the authors mention that **We also demonstrate that it is not necessary to include samples from every target category in the calibration dataset to get the representative activation ranges for quantization**, did the authors try to select the categories of data in their experiments according to a certain design rather than just randomly selecting them, and did the randomness of the data significantly affect the results of the experiments (judging from the authors' experimental setups, this would not be too much of a drain on resources, and therefore I think that obtaining the statistical significance of the methodology would be very necessary).
2. I'm very curious to see what are the differences between the author's method without access to BN information compared to DSQ, and what are the differences in the final simulated images by training under visualization; and what are the differences under the distribution of BN (even though the author's training doesn't use access to the BN, the ResNet network is still using the BN)
3. During the each training step, the author introduce additional noise to x (line 228), why? More explanation should be done for this.

---

### Official Review · Reviewer_hiCt · 2024-10-31

**Soundness:** 2
**Presentation:** 2
**Contribution:** 2
**Rating:** 5
**Confidence:** 4

**Summary:**

The article proposes a novel data-free approach for post-training quantization (PTQ) that does not rely on batch normalization (BN) statistics. Instead, it utilizes backpropagation to generate synthetic calibration datasets, making the approach applicable even to models without BN layers, such as AlexNet. The authors argue that it is unnecessary to use samples from every target class to create effective calibration data. They demonstrate significant improvements over state-of-the-art (SOTA) methods, especially in 4-bit quantization scenarios, with a particular focus on improving quantization efficiency with fewer synthetic samples.

**Strengths:**

(1) The article introduces an innovative method that circumvents the need for BN statistics in calibration data generation, broadening the scope of models that can benefit from post-training quantization.
(2) The authors provide an extensive set of experiments comparing their approach with SOTA methods. They use multiple network architectures and achieve consistent accuracy improvements, demonstrating the robustness of their method.
(3) The authors effectively show that only a small number of synthetic samples are needed for calibration, reducing the computational overhead without compromising accuracy. This is particularly useful for resource-constrained environments.
(4) Tables and figures are well-detailed, providing clear comparisons with other PTQ methods. This helps in understanding the practical benefits of the proposed method.

**Weaknesses:**

(1) The paper's writing, at times, becomes dense, particularly when discussing the methodology (e.g., Algorithm 1 and the synthetic data generation process). Simplifying some of these sections and using clearer language or more visual aids could help readers understand the core ideas more easily.
(2) Although the motivation behind avoiding BN statistics is well justified, the implications for practical deployment (e.g., specific hardware or real-world use cases) could be elaborated. This would help emphasize the real-world applicability of the proposed method.
(3) The paper does not sufficiently compare its data-free approach to non-data-free PTQ approaches that use real data. A more explicit comparison could help readers appreciate the trade-offs involved.
(4) The authors could benefit from adding more ablation studies on the impact of key hyperparameters, such as the number of iterations (T) or learning rate (α), to demonstrate their influence on quantization performance. This could provide a more complete understanding of how to fine-tune the approach in practice.
(5) The paper briefly acknowledges that BN-based calibration datasets can suffer from homogeneity, but a more thorough discussion on the limitations of the proposed method would make the paper more balanced. For example, scenarios where this approach might not work as effectively should be highlighted.
(6) The computational cost of generating the synthetic calibration dataset via backpropagation should be better discussed. The time complexity of this synthetic data generation could be compared to that of traditional calibration datasets.

**Questions:**

(1) Adding more visual illustrations, such as diagrams of the calibration dataset generation process, would make the methodology more accessible, especially for readers not well-versed in adversarial or synthetic data generation.
(2) Consider adding more use-case scenarios or case studies to illustrate where and how this method can be practically employed (e.g., edge devices in IoT, mobile devices, etc.).

---

> ### Comment · Reviewer_hiCt · 2024-11-26
>
> After reading the comments of other reviewers, I decide to lower my score.

---

### Official Review · Reviewer_Mcsj · 2024-11-04

**Soundness:** 2
**Presentation:** 2
**Contribution:** 2
**Rating:** 3
**Confidence:** 4

**Summary:**

This paper proposes generating synthetic data agnostic to batch normalization statistics in zero-shot quantization. Mixing Gaussian noise when generating synthetic data ensures robustness and diversity, and it experimentally demonstrates that generating samples for all categories is unnecessary.

**Strengths:**

1. The paper is easy to follow.
2. The proposed method can be easily applied to various DNNs.

**Weaknesses:**

1. The performance gain of the proposed method is marginal. In particular, data-free quantization methods have already performed excellently for 4-bit to 8-bit quantization. To clearly demonstrate the performance improvement of the proposed method, experiments should be conducted in low-bit (i.e., 2-bit or 3-bit) settings. For example, GENIE [1], a method that generates synthetic datasets for quantization, has shown strong performance in 2-bit and 3-bit quantization.
2. Since CIFAR-10 already has a small number of classes, it is somewhat insufficient to prove the claim in the paper that “selecting only a few target classes is sufficient.” Validation on datasets with a larger number of classes, such as CIFAR-100, is needed.
3. The proposed method claims performance gains regardless of the presence of batch normalization due to its agnostic nature. To support this, the method should be tested on a wider variety of models. While the paper only presents results for CNN-based models, it should also demonstrate performance on ViT-based models (e.g., ViT, DeiT and Swin Transformer).
4. The writing can be further polished. There is some inconsistency in terminology throughout the paper. The term “sample” needs to be clearly defined to indicate whether it refers to a category (i.e., class) or the number of synthetic data generated per category.

[1] Jeon et al., "GENIE: Show Me the Data for Quantization," CVPR 2023.

**Questions:**

Does the proposed method show performance improvements when applied to zero-shot quantization approaches other than the batch normalization statistics methods, such as latent embeddings (e.g., Qimera [2]) or text feature distribution (e.g., TexQ [3])?

[2] Choi et al, "Qimera: Data-free Quantization with Synthetic Boundary Supporting Samples," NeurIPS 2021.

[3] Chen et al., "TexQ: Zero-shot Network Quantization with Texture Feature Distribution Calibration," NeurIPS 2023.

---

### Official Review · Reviewer_72Qf · 2024-11-04

**Soundness:** 1
**Presentation:** 1
**Contribution:** 1
**Rating:** 1
**Confidence:** 4

**Summary:**

This paper propose a calibration dataset generation algorithm for PTQ, which is agnostic to BN statistics and leverage the back-propagation to create synthetic images. In addition, this paper demonstrate it is not neccesary to include samples from every target category in the calibration dataset.

**Strengths:**

1.  The synthetic image generation process without BN statistics supervision is helpful for networks without BN, also simplifying the optimization process.
2. This paper uses very few samples, no more than 35.

**Weaknesses:**

1. Existing works using BN statistics will fall back into this paper on No-BN networks, using only the label. This paper does not show its novelty.
2. More  comparison to the newest works should be applied, like [1], [2] and [3]

[1] Long-range zero-shot generative deep network quantization

[2] Hard Sample Matters a Lot in Zero-Shot Quantization

[3] IntraQ: Learning Synthetic Images with Intra-Class Heterogeneity for Zero-Shot Network Quantization

**Questions:**

what's the comparison to the newest works?

---

### Note · Authors · 2024-12-13

I have read and agree with the venue's withdrawal policy on behalf of myself and my co-authors.